# Therapeutic Potential of Bioactive Compounds from Edible Mushrooms to Attenuate SARS-CoV-2 Infection and Some Complications of Coronavirus Disease (COVID-19)

**DOI:** 10.3390/jof9090897

**Published:** 2023-08-31

**Authors:** Paran Baruah, Aparup Patra, Sagar Barge, Mojibur R. Khan, Ashis K. Mukherjee

**Affiliations:** 1Life Sciences Division, Institute of Advanced Study in Science and Technology, Paschim Boragaon, Garchuk, Guwahati 781035, Assam, India; paranbaruahaxom@gmail.com (P.B.); patra.aparup@gmail.com (A.P.); sagarbarge8@gmail.com (S.B.); mojibur.khan@gmail.com (M.R.K.); 2Faculty of Science, Academy of Scientific and Innovative Research (AcSIR), Ghaziabad 201002, Uttar Pradesh, India

**Keywords:** bioactive molecules anti-COVID-19, COVID-19 therapy, medicinal mushrooms, SARS-CoV-2, virus–host interaction

## Abstract

The novel severe acute respiratory syndrome coronavirus 2 (SARS-CoV-2), a highly infectious positive RNA virus, has spread from its epicenter to other countries with increased mortality and morbidity. Its expansion has hampered humankind’s social, economic, and health realms to a large extent. Globally, investigations are underway to understand the complex pathophysiology of coronavirus disease (COVID-19) induced by SARS-CoV-2. Though numerous therapeutic strategies have been introduced to combat COVID-19, none are fully proven or comprehensive, as several key issues and challenges remain unresolved. At present, natural products have gained significant momentum in treating metabolic disorders. Mushrooms have often proved to be the precursor of various therapeutic molecules or drug prototypes. The plentiful bioactive macromolecules in edible mushrooms, like polysaccharides, proteins, and other secondary metabolites (such as flavonoids, polyphenols, etc.), have been used to treat multiple diseases, including viral infections, by traditional healers and the medical fraternity. Some edible mushrooms with a high proportion of therapeutic molecules are known as medicinal mushrooms. In this review, an attempt has been made to highlight the exploration of bioactive molecules in mushrooms to combat the various pathophysiological complications of COVID-19. This review presents an in-depth and critical analysis of the current therapies against COVID-19 versus the potential of natural anti-infective, antiviral, anti-inflammatory, and antithrombotic products derived from a wide range of easily sourced mushrooms and their bioactive molecules.

## 1. Introduction

Towards the end of 2019, a few acute respiratory tract infection cases were reported in Wuhan, a prominent business hub in China. Subsequently, the outbreak was reported to the World Health Organization (WHO), and the causative agent was identified as a coronavirus. This unusual coronavirus displayed 96% and 79.6% genome similarity to a bat coronavirus and the severe acute respiratory syndrome coronavirus (SARS CoV), respectively [1,2]. The International Committee on Taxonomy of Viruses (ICTV) named this novel coronavirus SARS-CoV-2; the disease caused by this virus became COVID-19 in common parlance [3]. By mid-2022, the virus had successfully breached 220 countries and territories worldwide and consequently engendered the fourth and fifth global waves of COVID-19. The World Health Organization report released on 7 June 2023 confirmed that approximately 767,750,853 cases and a death toll of 6,941,095 had occurred worldwide and created an unprecedented global healthcare crisis (WHO report on 7 June 2023). All coronavirus (CoV) belongs to the order Nidovirales suborder Cornidovirineae and family Cornaviradae. This single-stranded RNA virus contains a genome size of about 26–32 kb, which is considered the second-largest known RNA genome [4,5].

As shown in Table 1, thirteen coronavirus variants have been discovered, and all are reported to be detrimental to humans [6]. Since the first discovery of coronavirus during the 1960s, SARS-CoV-2 is the seventh variant that has infected the human race and led to a global pandemic [7]. Therefore, based on the significant threat that the SARS-CoV-2 virus poses to public health, it has been designated a variant of concern (VOC). The virus also displays multiple phenotypes, can cause severe clinical manifestations, and employs various transmission strategies via nasal and vocal passages, physical contact, and virulence [8,9].

During the COVID-19 pandemic, the global market focused on bioactive ingredients that boost the immune system. Therefore, scientists worldwide resumed intensive studies on bioactive compounds that can boost the immune system to protect against SARS-CoV-2 and limit the accelerated transmission of this virus [10,11]. Consequently, bioactive compounds from herbal sources and edible mushrooms gained commercial interest due to their easy availability, high anti-oxidant activity, nutritional value, and low side effects.

There have been some comprehensive literature reviews on bioactive compounds from medicinal plants and herbs [12,13,14]; however, minimal studies have been conducted on the therapeutic activity of bioactive compounds derived from mushrooms against SARS-CoV-2 and other viruses. Most studies have focused on polysaccharides and polysaccharide–protein complexes, not on their secondary metabolites for treating viral infection [15,16]. Only a few reports are available on improving COVID-associated symptoms, such as lung infection, its effect on macrophages, and the thrombotic and cardiovascular effects of mushrooms [15,16,17,18,19,20,21].

This review article describes the potential anti-viral activity of edible mushrooms, specifically against SARS-CoV-2. In the beginning, we briefly describe the structure and infection mechanism of the SARS-CoV-2 virus, followed by the currently available treatments (besides vaccination) and their limitations for successful implementation. The main objective of this review is to critically analyze the roles and mechanisms of different mushroom-derived bioactive compounds in preventing SARS-CoV-2 infection and the pathophysiology associated with its infection, such as lung infection, inflammation, cytokine storm, and thrombotic and cardiovascular effects.

The literature review was performed based on articles published in significant databases, including Scopus, Google Scholar, PubMed, and ScienceDirect, using primary keywords such as “medicinal mushrooms”, “bioactive compounds from mushrooms”, “SARS-CoV2”, “medicinal mushrooms for viral therapy”, “viral infection”, and “SARS-CoV2”. Only those extracts or purified compounds from mushrooms with known mechanisms are considered. Reference lists linked to relevant papers were extensively examined for additional source material.

## 2. Structure and Mechanism of Infection

The SARS-CoV-2 genome comprises about ∼30,000 nucleotides that code for four structural proteins, viz., the innermost nucleocapsid (N) protein, the surface membrane (M) protein, the external envelope (E) protein, and the outermost spike (S) protein, along with several non-structural proteins (NSP) (Figure 1A). The innermost nucleocapsid (N) protein and the enclosed RNA strand act as viral factories after gaining entry into the cell aided by S, M, and E proteins [9]. The S protein mediates the entry of the virus into the host cell through fusion. In contrast, the M protein acts as a central organizer for the viral assembly, as it is the most abundantly found protein on the viral surface [3]. The E protein is a small viral membrane protein composed of ~76 to 109 amino acids, which helps permeate the membrane permeability of the host cell and viral assembly [10]. Hemagglutinin-esterase dimer (HE), found on the surface of the SARS-CoV-2 virus, is not required for replication but appears to be necessary for infecting the host cell by aiding entry [9]. The structure of SARS-CoV-2 is shown in Figure 1A. 

Infections occur via the nose and mouth mucosal membranes containing the receptor, angiotensin-converting enzyme 2 (ACE2), either by direct inhalation or indirect deposition from surfaces contaminated with respiratory droplets carrying the SARS-CoV-2 virus. ACE2 has recently been identified as the SARS-CoV-2 receptor, is best known for being the causative agent of mild upper respiratory tract infection, and is primarily expressed in vascular endothelium cells, kidneys, intestine, liver, heart, lungs, and testis [11,12]. The function of the binding of ACE2 is driven by endocytosis, leading to proteolytic cleavage and processing. ACE2 is the obligate receptor of SARS-CoV-2 and facilitates the entry of the virus into cells [13].

Once the virus enters the cytoplasm, it creates its own replication/transcription complex (RTC) to multiply its genetic material and gather new viral particles [11,12]. The positive-strand RNA genome of the virus is translated to generate viral replicate proteins. Viral replicate proteins use the genome as a template to create full-length negative-sense RNAs, which ultimately serve as templates to generate additional full-length genomes [14]. The replicated genomes form the nucleocapsids by the N protein in the cytoplasm, and finally, structural proteins and nucleocapsids result in self-assembly to produce new virions. Then, new virions are exported from infected cells, transported to the cell membrane, and secreted through exocytosis to infect the other cells [3,9]. The mechanism of SARS-CoV-2 infection in humans is illustrated in Figure 1B. 

When transmitted from human to human, the incubation period is four days. The earliest potential symptoms, like cough, fatigue, fever, and myalgia fever, occur within 14 days of exposure, and sometimes they appear earlier. A few of the earliest signs are a high fluctuating fever, dry cough, fatigue, muscle pain, headache, dyspnoea, and lymphopenia. A COVID-19-infected person faces difficulties with breathing and chest pain after five days of onset of the infection and acute respiratory distress syndrome (ARDS) after the seventh day [17].

## 3. Current Therapy against COVID-19: The Key Issues and Challenges

The difficulties created by the COVID-19 pandemic bolstered drug repurposing and focus on the treatment of COVID-19 and resulted in clinical trials for hundreds of antiviral drugs. Table 2 displays four mushroom-based drugs that have been approved for human trials, among which the trials for three are still in progress. It is noteworthy that COVID-19 patients are treated with mushroom-based products, compounds, and supplements. This section emphasizes the different drugs that are used to treat COVID-19. The selection criteria of these drugs include the putative mechanisms preventing the entry of viral particles and their replication in the host. Some of these drugs were previously used to treat SAR-CoV-2 infections, and a few are being used for the first time against SAR-CoV-2 disease [18]. The treatment strategies against COVID-19 can broadly be classified as using repurposed drugs, antibody-based immunotherapeutic methods, cell therapy, and nutraceutical supplementation.

### 3.1. Repurposed Drugs

Developing a new drug against a severe disease is a long, drawn-out, and expensive process, and thus, has a low success rate. Therefore, in the dire emergency created by the COVID-19 pandemic, most attempts were directed at identifying possible treatments from already extant approved or marketed drugs. Repurposing is one of the best approaches to organizing therapeutic options for global emergencies, such as COVID-19, in a limited time [19,20]. The main advantages of these types of drugs are the well-documented mechanism of action, pharmacokinetic and pharmacodynamics profile, and side effects. The concurrent disadvantages of repurposed drugs are their poor selectivity and feeble efficacy among unknown families of viruses [21]. Repurposed drugs, such as remdesivir (RDV), FDA-approved lopinavir/ritonavir (LPV/r), and hydroxychloroquine (HCQ), are widely used to treat severely affected COVID patients. However, SOLIDARITY trial studies published by the WHO showed that RDV, HCQ, and LPV/r had very moderate or no effect on the overall mortality rate, initiation of ventilation process, or duration of hospital stays of patients hospitalized with COVID-19 [18,22]. 

Some studies have also focused on repurposed drugs isolated from edible mushrooms. In silico analysis has revealed six compounds, namely colossolactone VIII, colossolactone E colossolactone G, ergosterol, heliantriol F, and velutin, as the best potential candidates for repurposed drugs for anti-SARS-CoV-2 activity; however, their exact action of mechanism is still unknown [23]. Among these compounds, ergosterol has been reported as an anti-inflammatory agent and has the potential as a successful therapeutic against SARS-CoV-2 [23]). In another study, four compounds, namely Kynapcin-12 (M_78), Kynapcin-28 (M_82), Kynapcin-24 (M_83), and Neonambiterphenyls-A (M_366), showed promising inhibitory activity in silico against the main protease (Mpro) of SARS-CoV-2, have been identified from the edible mushroom *Polyozellus multiplex*. Nevertheless, the in vitro and in vivo inhibitory activity of such compounds is yet to be revealed [24]. Therefore, more detailed in vitro and in vivo studies are warranted for using mushroom-derived bioactive compounds as therapeutic agents against SARS-CoV-2 and other viruses. 

### 3.2. Antibody-Based Immunotherapeutic Strategies

Based on the experience of clinicians treating other viral infections, antibody-based therapy gained popularity among COVID-19 patients [25]. Among different immunotherapy procedures, the administration of plasma from recovered patients into infected patients, called convalescent plasma (CP) therapy, was widely used. The presence of the neutralization of antibodies, immuno-modulatory cytokines, and autoantibodies in convalescent plasma were the leading therapeutic agents in CP. The impact and benefit of the use of CP as therapy against COVID-19 have been widely reported since the beginning of the pandemic [26]; however, the lack of established standard dosing methods, lack of the potent availability of donors, unknown host interactions, and severe adverse serum reactions have rendered CP therapy controversial and limited its applications [27]. In addition, the FDA has approved the emergency use of anti-virus monoclonal antibodies against the S protein of the virus, and studies have shown that this reduces the hospitalization rate by 70–85%. However, significant concerns are the high-cost manufacturing processes, labor intensity, lack of dose optimization, unknown individual-specific response, and adverse side effects.

### 3.3. Cell Therapy

With the progress of the pandemic, cell therapies based on stem cells, dendritic cells, natural killer cells, and engineered lymphocyte cells have gained prominence against COVID-19 [28]. Mesenchymal stem cell therapy has gained the most popularity among various cell-based therapies. Although clinical trial studies have demonstrated a significant reduction in mortality and hospitalization [29], the FDA has not approved stem cell-based therapies due to a lack of comprehensive data to estimate their therapeutic efficacy. Therefore, more research and randomized clinical trials are needed to establish the definitive mechanism for treating COVID-19 patients with stem cell-based therapy [30].

### 3.4. Nutraceutical Supplementation

Different nutraceutical supplements, like regular intake of vitamin C, vitamin D, zinc, and melatonin-containing supplements, have been recommended. The immunomodulatory properties of these molecules, such as boosting the immune system and a supportive role in COVID-19 patients, have been postulated for these supplements [31,32]. In this regard, edible mushrooms can be a rich source of vitamins and mineral supplements [33]. In the subsequent section of this review article, we discuss the mechanism and role of these to combat COVID-19. However, there are insufficient data to date to encourage or discourage the prescription of vitamins and mineral supplements for treating COVID-19 infections. More clinical trials are warranted to understand the role and mechanism of these nutraceutical supplements. 

## 4. Potential of Natural Products Derived from Mushrooms for the Treatment of SARS-CoV-2 Infection

Using artificial intelligence (AI), researchers have found a potential therapeutic agent that could inhibit viral infection and clathrin-mediated endocytosis, thus offering a potent therapy for COVID-19 [34]. The literature supports and encourages the use of products of natural origin to treat various viral diseases, including SARS-CoV and the Middle East respiratory syndrome coronavirus (MERS-CoV) [24]. The trial against SARS-CoV was initiated using mainly two mushrooms: turkey tail (*Trametes versicolor*) and agarikon (*Fomitopsis officinalis*). Mushrooms could probably act by immune-modulating against SARS-CoV-2. Taking a cue from the gut microbiome, mushrooms may boost the immune cells by binding T cells to the mushroom polysaccharides [35]. In addition, mushrooms possess a variety of bioactive polysaccharides or polysaccharide–protein complexes, which enhance innate and cell-mediated immune responses [36]. It is also well documented that mushrooms are a storehouse of medicinal properties, such as immunomodulating, antiviral, antibacterial, antifungal, cardiovascular, antihypercholesterolemic, antiparasitic, detoxification, antitumor, hepatoprotective, and antidiabetic effects. 

Natural substances and compounds extracted from mushrooms and herbs have potential antiviral and anti-inflammatory activities; thus, natural compounds can act as an abundant and potent source against COVID-19 [37]. The *Agaricomycota* among the *Basidiomycetes* mushrooms, *Agaricus subrufescens, Hericium erinaceus,* and *Grifola frondosa* are well-known and commonly used medicinal mushrooms that have been used worldwide as traditional medicine for treating a range of diseases. Basidiomycetes are gaining recognition for consumption based on the traditional knowledge that they can prevent cancer, give nutritional support during chemotherapy, and prevent chronic inflammatory conditions, like hepatitis and other related diseases. They contain bioactive compounds such as polyphenols, flavonoids, and other secondary metabolites. In addition to their anti-inflammatory properties, it is reported that Basidiomycetes mushrooms enhance the Th1 cellular immune response with an increase in the IFNɣ, IL-2, and IL-12 cytokines [34,35,36,38]. Generally, edible medicinal mushrooms are macroscopic fungi that mainly belong to the division of Basidiomycetes (oyster, *Agaricus*, *Pleurotus*, and many more) and sometimes to Ascomycetes (morels and truffles) [39,40]. These fruiting bodies can be found underground (hypogeous) or aboveground (epigeous) [37]. Since immemorial, these fungal members have grabbed attention as ultimate health foods [41]. 

### 4.1. Antiviral Activity of Mushroom-Derived Bioactive Compounds: Potential Therapeutics against SARS-CoV-2

The dire need to develop an antiviral product to inhibit viral infection has increased with the daily increase in human casualties from COVID-19. Polysaccharides, the macromolecular components in various mushroom species, have shown strong antiviral properties. The polysaccharide lentinan (LNT) isolated from *Lentinus edodes* (shiitake mushroom) has shown both direct inactivation and regulating immune responses via enhancing the functional activity of macrophages, mononuclear cells, and neutrophils. Therefore, lentinan (LNT) is named a biological response modifier (BRM). It consists of a β-(1 → 3)-glucan backbone with β-(1 → 6)-glucosyl as its side-branching units and is terminated by mannosyl or galactosyl residues, as shown in Figure 2 [42,43,44]. 

It has been observed that patients with severe SARS-CoV-2 infections are susceptible to *Herpes simplex* viral reinfection [28]. Therefore, to reduce the severity of COVID-19, patients co-infected with the herpes simplex virus must be treated first. The evaluation of water-insoluble β-glucans present in the sclerotia of *P. tuber-regium* along with corresponding water-soluble sulfated derivatives has shown to work against herpes simplex virus type 1 and type 2. This effect is achieved by binding sulfated products to viral particles, preventing infection in the host [29]. β-glucan, present in *Grifola frondose polysaccharide* (GFP) (maitake mushroom), is known for its substantial antiviral effects [30]. Furthermore, the extracted D-fraction from *G. frondose*, in unison with human interferon alpha-2b, showed a better response against HBV than individually [45]. When coupled with the HBV vaccine, *Poria cocos* polysaccharide-II from the *P. cocos* mushroom evoked a more robust immune response [46]. 

Lectins, the most studied carbohydrate-binding protein of mushrooms, have been employed in different therapeutic applications and even as a diagnostic tool in clinical practice [47,48]. Their unique molecular and physiological characteristics have not received adequate attention in antiviral property studies [49]. Lectins bind with high-mannose carbohydrates on viral envelope glycoproteins and entirely change the conformation arrangements vital for viral entry into the host body [50]. As suggested in the extant literature, the prime viral protease enzymes that are pivotal in the virus’s life cycle [51,52] are prime targets for drug development against SARS-CoV-2 [49,53].

In another study, a pair of two triterpenoids extracted from *Ganoderma lucidum* (reishi mushroom) demonstrated antiviral activity against enterovirus 71 without displaying cytotoxicity in human rhabdomyosarcoma cells [54]. Mushrooms are well known for their high protein content; one such protein is ubiquitin, which is found in oyster mushrooms and has been evaluated for rendering antiviral properties [55]. The isolation of five proteins from *Flammulina velutipes* (enoki mushroom)*,* when co-administered with HPV16 E7, enhanced the production of HPV-16 E7-specific interferon [56]. The aqueous extract of *Agaricus blazei* Murill, (AbM), administered in hepatitis B patients for one year, helped regulate liver functions [57]. Further, when AbM was administered in chronic HCV patients, it induced a stimulatory effect by releasing an immunological signaling substance in vitro. In contrast, in in vivo conditions, not many transformations in the levels of these substances were observed [58].

Influenza virus co-infection has also been reported to enhance COVID-19 severity among COVID-19-infected patients. The similarity in respiratory symptoms makes distinctions between signs of influenza virus infection and COVID-19 difficult [53]. Dietary interventions with mushrooms have been shown to reduce flu-like symptoms caused by the influenza virus. Consequently, CO_2_ extracts obtained from the members of the mushroom family, like *Lentinus edodes*, *Trametes versicolor*, *Ganoderma lucidum*, *Flammulina velutipes*, *Pleurotus eryngii*, *Pleurotus ostreatus*, *Fomes fomentarius*, *Lyophyllum shimeji*, and *Schizophyllum commune*, worked positively in inhibiting the spread of influenza A H1N1 virus in varying titers of Madin–Darby canine kidney cells [59]. A study demonstrated promising results with mushrooms like *Trametes gibbosa*, *Trametes Versicolor*, *Ischnoderma benzoinum*, *Lenzites betulina*, *Laricifomes officinalis*, and *Daedaleopsis confragosa* when used for treatment against influenza-type viruses H5N1 and H3N2, showing they were accountable for reducing inflammation, congestion, and epithelial necrosis of the larger airways [60]. 

Hexose-related compounds extracted from Basidiomycetes mushrooms successfully reduced the mortality rate of C57 black 6 (C57BL/6) mice infected with the influenza virus [50]. Sesquiterpenoid isolated from *Phellinus igniarius* (willow bracket mushroom), has been reported to inhibit the neuraminidase activity of influenza virus [51]. Compounds like applanoxidic acid G, lucialdehyde B, lucidadiol, and ergosta-7, 22-diene-3b-ol (Figure 3A–E), isolated from *Ganoderma pfeifferi*, have also been documented to work against influenza virus [52]. Another study reported the inhibitory activity of the aqueous and ethanolic extracts of *Agaricus brasiliensis* (almond mushroom) against poliovirus type 1 (PV1) using HEp-2 cells. The maximum viral inhibition of the ethanolic extract and aqueous extracts were 58.47% and 14%, at concentrations of 1000 µg/mL and 200 µg/mL, respectively [49]. Polysaccharides and aqueous and ethanol extracts obtained from *Lentinula edodes* inhibited the replication of poliovirus type 1 [53]. 

Encephalomyocarditis virus (EMCV) and coronavirus activate the same inflammatory pathway in infected patients via activation of the NLRP3 inflammasome [61]. A study elaborated on the activity of the endopolysaccharides and laccase present in *Cerrena unicolor* against both human herpesvirus type- 1 (HHV-1) and encephalomyocarditis virus (EMCV) using the SiHa and L929 cell lines [62]. The results demonstrated that laccase at concentrations of 0.025 µg/mL and 0.25 µg/mL for the cell lines SiHa and L929 were the most active against HHV at an early stage of viral replication, with a resulting decrease in the virus titer by 1.64 and 2.1 logs, respectively. 

### 4.2. Mushroom-Derived Anti-Inflammatory Compound: Potential Candidate to Reduce COVID-19-Associated Inflammation

In most cases of SARS-CoV-2, the dual impact of hyper-inflammation of the lungs and severe respiratory disorder has been recorded. The most critical condition associated with SARS-CoV-2 infection is the end step of hyperinflammatory reactions, referred to as hyper-cytokinemia or a “cytokine storm” [63]. Complete insight into the immunopathogenesis of the cytokine storm in COVID-19 underscores the reduction of morbidity risk by prospects of detection and early-stage remission [64]. 

#### 4.2.1. Inflammation and Immune System

The inflammatory phase of COVID-19 is characterized by an excessive decrease in the lymphocyte count offset by an abnormal increase in the C reactive protein (CRP). Particular subsets of T lymphocytes cells (CD3+ and CD4+ T cells and CD3+ and CD8+ T cells) are reduced during the COVID-19 infection period and reach significantly lower levels in severe cases [58]. 

It was found that inflammatory responses varied and depended on the virus’s geographical variant circulating in the population [65,66]. Therefore, elevated levels of IL-6 were mainly detected in all studies, whereas an increase in IL-10 was reported in only one study [67]. Another study showed that patients with difficult or critical situations had higher levels of G-CSF, GM-CSF, IP-10, MCP-1, MIP-1a, MIP-1b, RANTES, and IL-8.

Inflammatory responses activate the immune system, and a robust immune system is a powerful weapon to tackle the novel COVID-19 pathogen. Therefore, bioactive components that strengthen the immune system and/or enhance inflammatory activities are strong contenders for SARS virus prophylaxis. As shown in Figure 4A–F, compounds like Colossolactone G, Heliantriol F, Ergosterol, Colossolactone VIII, Velutin, and Colossolactone E are found in various mushroom species such as *Agaricus bisporus*, *Flammulina velutipes*, *Ganoderma lucidum*, *Laetiporus sulphureus*, *Lentinus lepideus*, *Leucoagaricus leucothites*, *Macrocybe gigantean*, *Pleurotus ostreatus*, and *Lentinula edodes*, and have already been studied for their anti-inflammatory properties [23]. The intracellular and extracellular proteins found in *Pleurotus ostreatus* have been shown to have immuno-modulating effects [68]. 

#### 4.2.2. Lung Infection and Inflammation in COVID-19

An uncontrolled delivery of pro-inflammatory cytokines causes a challenging inflammatory or cytokine storm (CS) in the lungs of patients suffering from acute COVID-19 (Figure 5) [69]. β-glucans are a naturally occurring group of β-D-glucose polysaccharides in Basidiomycota, exhibiting a strong binding affinity to the immune cell surface receptors via pattern recognition receptors (PRRs) as pathogen-associated molecular patterns (PAMPs) [70]. Thus, in severe COVID-19 cases, a reliable source of β-glucan can significantly reduce the cytokine levels involved in cytokine storms (Figure 5). Further, β-glucans derived from the mycelia of *Lentinus edodes* showed good potential for recovery. *L. edodes* contains higher levels of β-glucan and lower levels of α-glucan. 

However, both β-glucan and α-glucan performed immunomodulatory activities and lowered the rate of inflammation in a lung epithelial model at a dose of 1 mg/mL. Furthermore, commercial (Carbosynth–Lentinan, CL) lentinan extract containing a lower amount of β-glucan and a higher amount of α-glucan showed efficacy against pulmonary cardioprotective effects and in vitro immunomodulatory effects in human alveolar epithelial A549 cells. Considering the above, β-Glucan from *Lentinus edodes* has considerable potential and effectiveness for improving and treating lung damage. The study showed that lentinan products had potentially reduced inflammation at a lower dose [71]. 

Another mushroom species, *Trametes versicolor* (common polypore mushroom) contains the compound (1→6)-ß-D-glucans as an active ingredient, which stimulates the immune system, thereby increasing the survival rate among colorectal cancer patients. (1→6)-β-D-glucans obtained from species like *Inonotus obliquus* (Chaga mushroom) and *A. bisporus* (Portobello mushroom) also displayed anti-inflammatory properties [72]. Many glucans like schizophyllan from *Schizophyllum commune*, grifolan from *Grifola frondosa*, and sclerotinia sclerotiorum glucan (SSG) from *Sclerotinia sclerotiorum* have anti-carcinogenic properties, attributed to the synchronization of both innate and adaptive immunity [73,74]. 

The two putative receptor proteins, transmembrane serine protease 2 (TMPRSS2) and transmembrane ACE2, that permit SARS-CoV-2 entry into host cells [75] were characterized as an androgen-regulated gene present in the prostate gland [76,77]. However, recent studies have revealed it to be positively regulated by androgen in human lung cells [78]. Findings showed that the inhibition of the androgen signaling axis impacted the TMPRSS2 expression in organs like the lungs and prostate gland [79]. *Agaricus bisporus* (white button mushroom) extracts could terminate TMPRSS2 expression by anti-androgenic activity via conjugated linoleic acid and by initiating an anti-inflammatory reaction, possibly through β-glucans. Further, in vivo, pre-clinical toxicity studies on eight-week-old male intact C57BL mice at (The Jackson Laboratory, Sacramento, CA) showed no toxicity of *Agaricus bisporus* (white button mushroom) [80,81]. 

#### 4.2.3. Macrophages and Inflammation in COVID-19

In mushrooms, anti-inflammatory activities were also analyzed using macrophage cell lines activated by stimulants like cytokines, stress, or bacterial components, which produced mediators that play a crucial role in many inflammatory diseases [82]. The involvement of these mediators in several diseases, including SARS-CoV-2, lowers their specificity in anti-inflammatory applications by suppressing the number of mediators in activated macrophages, which may help stop the disease. Some mushroom species are known to interrupt the expression of these inflammatory enzymes or mediators, such as nitric oxide synthase (iNOS) and COX-2 (cyclooxygenase enzymes), which are envisaged as potential candidates for treating inflammatory disorders in COVID-19. Similarly, there is a growing possibility of using COX enzymes as new anti-inflammatory drug prototypes. 

The extracts of medicinal mushrooms like *Armillariella mellea* or *Antrodia camphorata* performed well against inflammatory reactions in a macrophage culture by inhibiting cytokines, nitric acid (NO) production, and PGE2, COX-2, and iNOS expression in response to lipopolysaccharides (LPS) [83]. The reishi mushroom (*Ganoderma lucidum)* is a promising inhibitor in the mRNA expression of inducible nitric oxide synthase (iNOS), followed by the reduction in lipopolysaccharide (LPS)-induced nitric acid (NO) production in macrophages [84]. An evaluation of *Inonotus obliquus* methanolic extract revealed it to be a significant inhibitor of inflammatory mediators, such as NO and PGE2, by inhibiting the expressions of iNOS mRNA and COX-2 mRNA, and therefore, suggested its therapeutic use for immunomodulation and anti-inflammation [75]. The n-butanol fraction of ethanolic extract of *Phellinus linteus* displayed a significant role in anti-inflammatory activity in the macrophage cell line RAW 264.7 [85]. 

When extracted with methanol, *Pleurotus florida*, an edible mushroom, showed considerable activity in mitigating carrageenan-induced inflammation and chronic inflammation by formalin [86]. It also showed prominent platelet aggregation-inhibiting activity, which encourages potential therapeutic applications for vascular disorders [87]. 

Polysaccharides obtained from *Pholiota nameko* (PNPS-1) were reported to show anti-inflammatory effects in various models of inflammation, like acute exudative, subacute, and chronic proliferative inflammation, without any ulcerogenic activity. Though the polysaccharide was not identified in this study, the data suggest a potential agent against different inflammatory-related diseases [88]. The intake of *Agaricus bisporus* substantially enhanced secretory immunoglobulin A (IgA) secretion in healthy human volunteers, displaying latent fitness properties for boosting mucosal immunity [89]. 

In certain situations, in the treatment for SARS-CoV, the inclusion of minimal quantities of zinc and pyrithione has had a powerful impact. Thus, RNA virus replication weakened when pyrithione and zinc ionophores were intracellularly supplemented. Consequently, dietary zinc supplements may impact COVID-19-related symptoms like diarrhea, lower respiratory tract infection, and COVID-19. There are reports of various species, like *Agaricus bisporus*, that are valuable sources of supplements and contain a significant amount of absorbable minerals, like K, Fe, Zn, Cu, Ca, Na, Se, Mn, and Mg [90].

One of the extensively studied extracts from cultured *Lentinula edodes* mycelia, an active hexose-correlated compound (AHCC), is composed of 74% of the dry weight of polysaccharides, of which 20% is a partially acetylated α-1,4-glucan. The intake of AHCC is widely reported to improve human health significantly and animal immunity against various viral diseases by modulating natural killer T (NK-T) cells, and gamma delta T (γδ-T) cells [91]. 

Further, coprinoid mushrooms possess bioactive molecules, like sterols, sesquiterpenes (cuparane, illudins), polysaccharides, proteins (hydrophobins and galectins), quinones (5-methoxy-p-toluquinone, benzoquinone, lagopodins, and hydroxylagopodins), volatile (skatole) compounds, derivatives of imidazole, and indolic compounds (triptamine, serotonin, bufotenin, psilocin, psilocibin, and ergothionin). These molecules deliver strong antifungal, antibacterial, antiviral, fibrino- and thrombolytic, neuro, and vasotocin effects [92]. 

### 4.3. Antithrombotic Effects of Bioactive Compounds of Edible Mushrooms

Within a short time, the cases of COVID-19 skyrocketed, and reports of the high frequency of thrombotic complexities in COVID-19 patients were disseminated widely. The consequence of the pathophysiology of COVID-19 is not clear at present, but the probable mechanism is considered to be systemic inflammation. In addition, coagulopathy is one of the outcomes that leads to a high risk of venous thromboembolism [93]. Other findings among patients also support the symptoms of microvascular and macrovascular thrombotic complications, including arterial and predominantly venous thromboembolism (VTE) [94,95,96,97].

Elevated D-dimer levels are also associated with COVID-19 infection, and its testing is a recommended laboratory procedure for investigating venous thromboembolism [58,98,99]. Thrombo-inflammatory biomarkers have been backed by numerous clinical trials concerning weak prognosis in patients with COVID-19 infection. These biomarkers include fibrinogen and D-dimer levels, prothrombin time (PT), and activated partial thromboplastin time (aPTT) [58,65,100].

Thrombosis is a complex blood clot process of one of two types—arterial or venous thrombosis. Platelet aggregation and coagulation cascade are the two main processes of thrombosis. In the case of platelet aggregation, both collagen and adenosine diphosphate (ADP) play pivotal roles as an agonist. Several countries and clinicians have already adopted COVID-19-related thrombosis treatments, and it is noteworthy that several other countries are expected to prepare similar thrombosis guidelines soon. Some commercial drugs are recommended for the treatment of COVID-19-associated thrombosis, categorized as blood thinner anticoagulants (argatroban—direct thrombin inhibitor, heparin—direct FXa inhibitor); antiplatelet drugs (abciximab—vitamin K antagonist); phosphodiesterase inhibitor (dipyridamole); and clot-bursting thrombolytic drugs (plasminogen activator). These drugs are handicapped by adverse reactions, drug–drug interactions, and a lack of detailed studies on COVID-19-related thrombosis [90]. In this context, mushroom extracts and natural compounds from mushrooms may be proposed as efficacious treatments of choice for COVID-19-associated thrombosis; nevertheless, intensive research and proper scientific validation are warranted. 

Various mushrooms have been evaluated for the treatment and prevention of thrombosis [101]. The property of dose-dependent inhibition in *Cordyceps militaris* (CMEE) ethanol extract on ADP- and collagen-induced platelet aggregation had no effect on the coagulation time. The relevant data display significant antithrombotic activities via antiplatelet rather than anticoagulation activity. *C. militaris* contains a bioactive component responsible for human platelet aggregation inhibition [102]. Reports show that cordycepin (Figure 6A) can significantly enhance cellular cAMP and cGMP levels, thereby inhibiting intracellular Ca^2+^ and TXA2 production without adversely affecting PLC-γ2 or IP3 [102]. 

Cordycepin-enriched (Figure 6A) WIB801C impeded ADP-induced platelet aggregation with an IC_50_ value of 18.5 μg/mL. The probable mechanism involved cAMP elevation, which aids in IP3RI (Ser1756) phosphorylation by CEWIB801C, which in turn hinders Ca^2+^ mobilization and VASP (Ser157) phosphorylation to interrupt αIIb/β3 activation [103]. Oligoporin A (Figure 6B) obtained from *Oligoporus tephroleucus*, an edible mushroom, hindered the collagen-induced platelet aggregation but showed no impact on ADP- and THR-induced platelet aggregation. In addition, its possible role in increasing cellular cAMP and cGMP in platelets has also been predicted [104]

The composition of amino acid in the inhibitor of platelet aggregation was identified as Gly (50%), Cys (25%), and Typ (25%), and the amino acid sequence was Trp-Gly-Cys. *Inonotus obliquus* was purified, and its chemically synthesized peptide (W-G-C) enabled increased platelet aggregation of 83.3% in an in vivo mice model [105]. 

A novel fibrinolytic enzyme, ACase, obtained from the mushroom *Agrocybe aegerita*, sequentially degraded the Aα band, followed by Bβ and γ chains of fibrinogen responsible for clotting [106]. ACase not only played dual roles of a plasmin-like protein and a plasminogen activator but also revealed the ability to deteriorate human serum albumin (HAS) and human immunoglobulin IgG.

Furthermore, thrombin, vital for the thrombosis process, was dismantled by ACase, to a considerable extent [107]. Fibrinolytic properties have also been reported in mushrooms like *Armillaria mellea*, *Pleurotus ostreatus*, *Flammulina velutipes*, *Grifola frondosa*, and *Tricholoma saponaceum*. *Pleurotus ferulae* (king oyster mushroom) is a rich source of active proteins and compounds, and the antithrombotic fibrinolytic metalloprotease is also claimed to have been obtained from it. The antithrombotic fibrinolytic metalloprotease displayed significant activity against the two critical factors needed for fibrin clot formation, i.e., thrombin and fibrinogen, and consequently disrupted the coagulation pathway. These results imply the antithrombotic nature of the enzyme by mediating the degradation of both pro-coagulant protease and fibrinogen by an interaction of fibrinogen and thrombin. It also includes the lysis of the fibrin polymer directly. If consumed, it will benefit not only as a dietary supplement but also as a therapeutic agent acting against thrombotic disorders [108]. In the fruiting bodies of *Pleurotus Sajor-caju*, two fibrinolytic protease enzymes, namely FP I and FP II, have been reported. Their purified form eliminated the β and γ chains of human fibrinogen. The substantial hindrance to the enzyme generated by 1,10 phenanthroline and EDTA has suggested the presence of a metalloprotease [109].

The fibrinolytic and antithrombotic roles of *Ganoderma lucidum* through its Zn^2+^ metalloprotease enzyme have been reported. In carrying out the fibrin plate method with its extract, the enzyme hydrolyzed both human fibrin and fibrinogen and exhibited prominent anticoagulant activity. It was able to degrade the α- and β-chains but did not hamper the γ-chain of the human fibrinogen. The same enzyme was also used to detect the anticoagulant activity using the thrombin time (TT) and activated partial thromboplastin time (APTT) tests in human plasma. It must be mentioned that it extended both the thrombin time (TT) and activated partial thromboplastin time but was enzyme concentration-dependent. *Armillariella mellea*, also known as honey mushroom, was reported to have a fibrinolytic metalloprotease enzyme, which, post-incubation with fibrinogen, preferentially degraded its α and β chains [110]. However, the γ chain remained viable for around 1.5 h before degrading. As a metalloprotease enzyme, its activation pathway for acting on fibrin clots might be distinct. Despite its broad-spectrum nature, the enzyme broke down only fibrinogen and fibrin; while other human plasma proteins, like thrombin, urokinase, immunoglobulin G human serum albumin, and hemoglobin, were not impacted. [111]. The existing literature also supports the presence of fibrinolytic enzymes in some more members of the mushroom family, like *Tremella fuciformis* and *Hericium erinaceum*.

At this juncture, it is noteworthy that the degree of degradation of these fibrinogen chains by the enzymes is species-dependent [112]. There are reports of two metalloendopeptidases, namely TSMEP1 and TSMEP 2, in the fruiting bodies of a wild mushroom called *Tricholoma saponeum*, which rendered fibrinolytic effects. This fact is backed by the reports of its ability to break the α and β chains of fibrinogen [113]. 

*Pleurotus ostreatus*, a white-rot basidiomycete, played a fibrinogenolytic role with the help of an enzyme isolated from it. On SDS-PAGE, the fibrinogen degradation by the purified enzyme was determined and found to break the α chain initially succeeded by the β and γ chains of fibrinogen. However, the deterioration time varied for each chain, i.e., 3 min for α, 45 min for β, and 10 h for the γ chain. The breakdown occurred without fibrin formation, which indicates that the cleavage sites of both α and β chains were unlike those of thrombin [114]. 

Wulfase, a fibrinolytic protease isolated from the medicinal mushroom *Sparassis crispa* Wulf. ex. Fr., showed numerous prohibitory actions against thrombin and factor Xa activities. For understanding the anticoagulant properties of this protease, its effect on APTT and PT on human plasma was determined and found to be remarkably delayed due to wulfase. A delay in APTT indicated the interruption of the intrinsic and common pathways, while the extension of PT implied the repression of the extrinsic pathway of the coagulation cascade. Furthermore, the fibrin-agarose plate method was performed to determine the skill of wulfase in fibrin clot lysis and prominent fibrinolytic activity in the fibrin plate was evinced. A circle was formed in the fibrin plate, and its size coincided with that developed by plasmin. These results endorse wulfase as a strong candidate to obstruct the vital steps in the pathogenesis of thrombosis [115]. The bioactive compounds isolated from different mushrooms and their therapeutic roles are summarized in Table 3. 

## 5. Future Scope and Conclusions

The spread of SARS-CoV-2 infection has made humankind aware of the limitations of knowledge in virology. The treatment or cure for this virus is challenging, as it has globally put a risk on the human population. Many currently introduced drugs to deal with this pathogen are based on emergency or clinical trials, excluding their side effects. Further, the constantly altering recommendations indicate the sensitivity of the issue. Antiviral drugs introduced to tackle the crises are associated with side effects, like pancreatitis, gastrointestinal events, and hepatic injury risk, to mention just a few. Further recommended antithrombotic drugs, like heparin, aspirin, and warfarin, are at risk of complications caused by bleeding [102,122].

Considering all these phenomena, searching for or investigating a drug of biological origin with minimal side effects and high efficacy is of the utmost concern and urgency. Bringing mushrooms into the picture engenders a reliable approach, as their derivatives are often valuable tools in health and disease because of their immunity prospects. They are in use around the globe as dietary supplements and functional foods, and therefore, problems associated with immune dysregulation and, more importantly, respiratory disorders are well documented [123]. The multifaceted roles played by these Basidiomycetes members of the fungi kingdom reverberates with the words of Hippocrates *“Let food be your medicine and medicine be your food”* [106]. 

Their quick natural growth and simple mode of reproduction in culture conditions are helpful in the biotechnological field for obtaining desired bioactive molecules and biotech products [94]. Over time, innovative methods have been developed that are helping humankind to secure an enhanced comprehension of these fungi kingdom members. Thus, a conclusion can be drawn that, if employed suitably, mushrooms with prospective anti-COVID-19 constituents will undoubtedly lead humankind to bring about efficient products to combat this novel SARS virus. 

## Figures and Tables

**Figure 1 jof-09-00897-f001:**
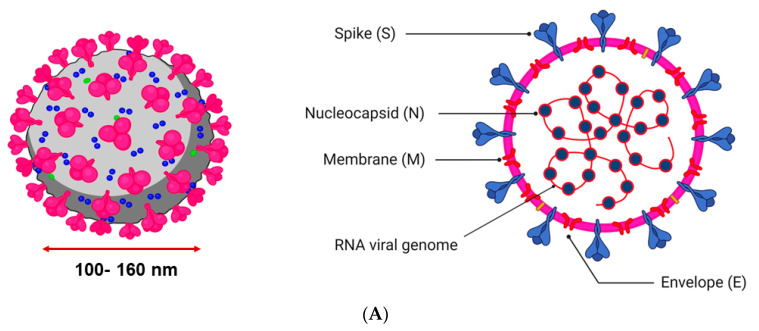
(**A**). Structure of SARS-CoV-2. This figure was created in BioRender with a paid subscription (agreement number: FU25P9V166). (**B**). Schematic diagram showing the mechanism of SARS-CoV-2 infection. The process starts with the attachment of the viral spike (S) protein to the host angiotensin converting enzyme 2 (ACE2) receptor and the facilitation of cellular entry by host cell protease (TMPRSS2). The endocytosed SARS-CoV-2 releases the virus into the host cell, and viral RNA is translated into viral replicase. The replicase polyproteins are subsequently cleaved into non-structural proteins (NSPs). The enzyme replicase uses a positive-strand RNA genome template to generate full-length negative-strand RNA. The RNA-dependent RNA polymerase (RdRp) transcribes a series of sub-genomic mRNAs that are translated into viral structural proteins (spike (S), envelope (E), nucleocapsid (N), membrane (M), and accessory protein (A)). The positive-strand RNA and viral proteins are assembled in the endoplasmic reticulum (ER) and Golgi. The newly formed virus particles are released from the infected cells by exocytosis and are ready to infect other healthy cells. This figure was created in BioRender with a paid subscription (agreement number: HR25PB901A).

**Figure 2 jof-09-00897-f002:**
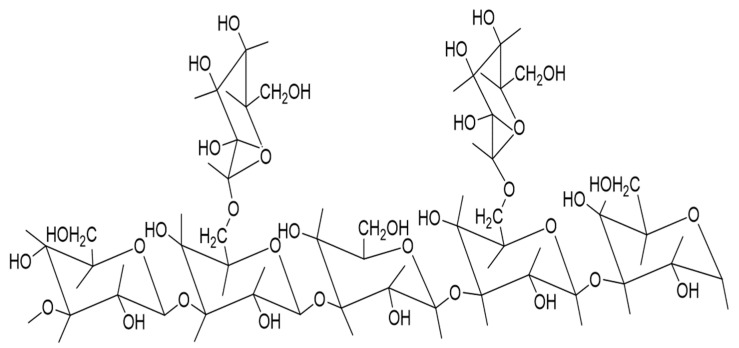
Structure of β-(1 → 3)-glucan backbones with β-(1 → 6)-glucosyl side-branching units.

**Figure 3 jof-09-00897-f003:**
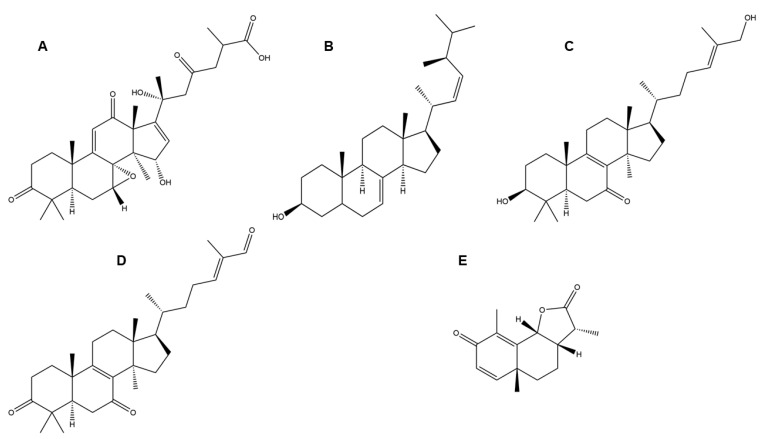
Structure of (**A**) applanoxidic acid G, (**B**) ergosta-7, 22-diene-3b-ol, (**C**) lucidadiol, (**D**) lucialdehyde B, (**E**) sesquiterpenoid.

**Figure 4 jof-09-00897-f004:**
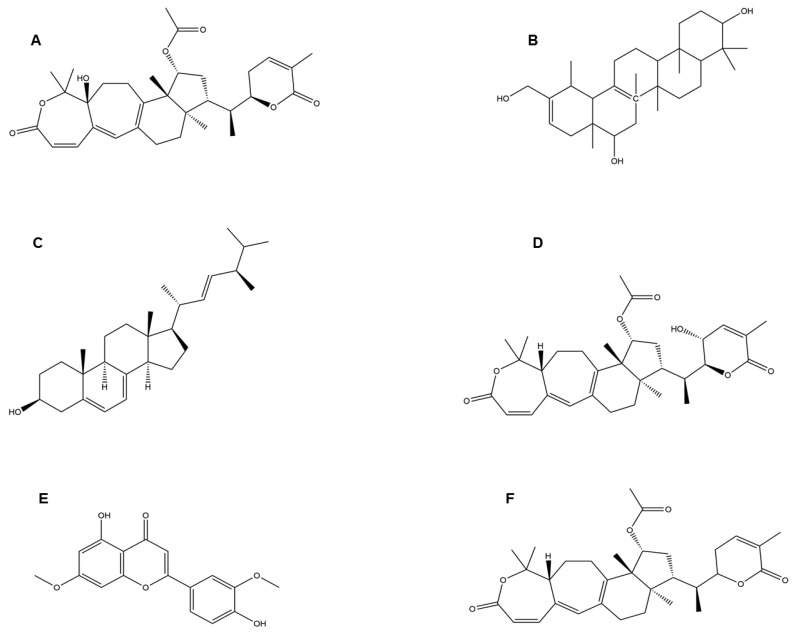
Structure of (**A**) colossolactone G, (**B**) heliantriol F, (**C**) ergosterol, (**D**) colossolactone VIII, (**E**) velutin, and (**F**) colossolactone E.

**Figure 5 jof-09-00897-f005:**
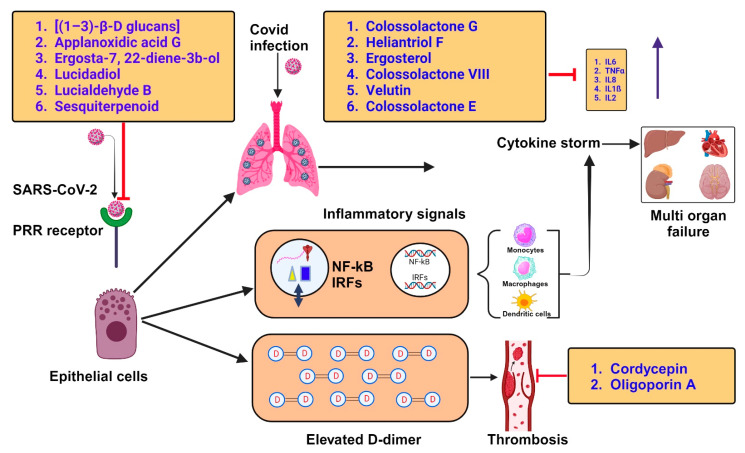
Patients suffering from acute COVID-19 showed an inflammatory or cytokine storm (CS) in the lung with an immoderately uncontrolled delivery of pro-inflammatory cytokines and the effect of naturally derived compounds during inflammation. This figure was created in BioRender with a paid subscription (agreement number: DH25PFBESY).

**Figure 6 jof-09-00897-f006:**
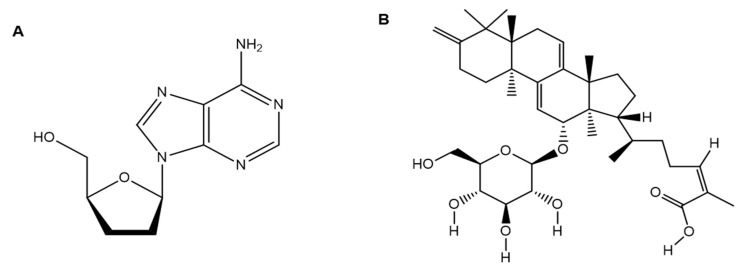
Structure of (**A**) cordycepin and (**B**) oligoporin A.

**Table 1 jof-09-00897-t001:** Schematic, algorithmic representation of the SARS-CoV-2 variant landscape [6]. The updated list of strains and lineage can be found at this link: https://www.who.int/activities/tracking-SARS-CoV-2-variants (accessed on 21 February 2022).

Serial No.	Strain	Lineage
1.	Alpha	B.1.1.7
2.	Beta	B.1.351
3.	Gamma	P.1
4.	Delta	B.1.617.7-3
5.	Epsilon	B.1.427-429
6.	Zeta	P.2
7.	Eta	B.1.525
8.	Theta	P.3
9.	Iota	B.1.526
10.	Kappa	B.1.617.1
11.	Lambda	C.37
12.	Omicron	B1.1.529
13.	Omicron	B.1.1.529

**Table 2 jof-09-00897-t002:** Summary of clinical trials with mushroom-based drugs against COVID-19-infected individuals. The information was sourced from Clinical Trials.gov (https://www.clinicaltrials.gov/ct2/results?cond=covid&term=Mushroom&cntry=&state=&city=&dist= accessed on 19 June 2022).

Serial No.	Title	Conditions	Intervention	Location
1.	*RCT of Mushroom-Based Natural Product to Enhance* *Immune Response to COVID-19 Vaccination*	COVID-19 Vaccination	Dietary supplement	University of California, San Diego, United States
2.	*Mushroom-Based Product for COVID-19*	COVID-19	Drug: Fo Tv	University of California, Los Angeles Los Angeles, California, United StatesUniversity of California, San Diego San Diego, California, United States
3.	*COVID-19: Collecting Measurements of Renin–Angiotensin System Markers, such as Angiotensin-2 and Angiotensin 1–7 (Tomeka)*	COVID-19	Combination product: Tomeka^®^ Drug: “Vernonia amygdalina”	Cliniques Universitaires de Kinshasa Kinshasa, Congo, The Democratic Republic of the Congo
4.	*Vitamin D3 Supplementation in Patients with Serum Values +/− 20 ng/mL*	COVID-19 Influenza A Influenza B H1N1 Influenza	Dietary supplement: Vitamin D3 supplementation Dietary supplement: Diet and sun exposure	Hospital Clinica Nova de MonterreySan Nicolás De Los Garza, Nuevo León, Mexico

**Table 3 jof-09-00897-t003:** Summary of the list of bioactive compounds isolated from mushrooms and their reported therapeutic activity and mechanism of action against viral infections.

Serial No.	Mushroom Species (Common Name)	Functional Molecules	Function	Reference
1.	*Lentinus edodes* (Shiitake mushroom)	β-Glucan	Most beta-glucans tested exhibited immunomodulatory activity by binding to receptors like dectin-1, toll-like receptors (TLRs), complement receptors type 3 (CR3), scavenger receptors (Src), and lactosylceramide receptors (LacCer) on immune cells. Thus, β-glucan can protect altered immune responses against various viral infections.	[116]
		Lentinan (LNT)	Lentinan extracts reduced cytokine-induced NF-κB activation in human alveolar epithelial A549 cells and effectively attenuated pro-inflammatory cytokine production (TNF-α, IL-8, IL2, IL-6, IL-22) as well as TGF-β and IL-10. It attenuated oxidative stress-induced early apoptosis, and thus showed in vitro immunomodulatory and pulmonary cytoprotective effects that may also have positive relevance to candidate COVID-19 therapeutics targeting cytokine storm.	[71]
2.	*Ganoderma pfeifferi* (Beeswax bracket)	Applanoxidic acid G Ergosta-7, 22-diene-3b-o l Lucidadiol Lucialdehyde B	Ganoderma triterpenoids inhibited influenza A H1N1 (A/WSN/33) infection by its antioxidant activity. It may modulate immune responses by binding to receptors like toll-like receptors (TLRs) and affecting the production of inflammatory cytokines. By doing so, Ergosta-7,22-diene-3b-ol could help regulate immune responses and reduce excessive inflammation associated with respiratory conditions. Through its anti-inflammatory and immunomodulatory activities, Lucidadiol has shown promising antiviral activities in preclinical and in vitro studies. This compound has shown antiviral activities against several viruses, including influenza A, HIV-1, HSV-1, HSV-2, HPV, HBV, and EV71. These antiviral effects are attributed to their ability to interfere with viral replication and entry into host cells. Specific antiviral mechanisms include inhibition of viral enzyme activity, interference with viral attachment to cellular receptors, and modulation of host immune responses to combat viral infection.	[2] [117] [118] [118]
3.	*Phellinus igniarius* (Willow bracket mushroom)	Sesquiterpenoid	This compound has demonstrated anti-influenza activity by inhibiting the Neuraminidase (NA), a viral surface protein, and thus, can be used as a potent antiviral drug.	[2,51]
4.	*Agaricus bisporus* (Button mushroom), *Flammulina velutipes* (Velvet foot; winter mushroom), *Ganoderma lucidum* (Lingzhi mushroom), *Laetiporus sulphureus* (Sulphur polypore), *Lentinus lepideus* (Scaly sawgill), *Leucoagaricus leucothites* (White dapperling), *Macrocybe gigantean* (Boro dhoodh chhatu), *Pleurotus ostreatus* (Oyster mushroom)	Colossolactone G Heliantriol F Ergosterol Colossolactone VIII Velutin	This compound has shown anti-viral activity against HIV and other viruses by inhibiting the surface proteins’ protease activity, such as HIV-1 protease. Heliantriol F exhibits characteristics as a potential inhibitor of the SARS-CoV-2 main protease (Mpro) and a rapid capturer of coronaviruses by strongly binding to the ACE2 receptor binding domain. Ergosterol shows anti-viral activity against a broad range of viruses by anti-inflammatory actions, reducing oxidative stress and immunomodulatory activities. This compound is reported to inhibit the SARS-CoV-2 main protease (Mpro) and rapidly captures coronaviruses by strongly binding to the ACE2 receptor binding domain, thereby preventing the entry of the virus into the respiratory tract. Velutin is shown to halt the virus’s protein synthesis and inhibit reverse transcriptase activities, thus restricting viral proliferation.	[119] [23] [120] [121]
5.	*Cordyceps militaris* (Scarlet caterpillar club fungus)	Cordycepin	Cordycepin showed inhibitory affinities against the principal SARS-CoV-2 protein targets (e.g., SARS-CoV-2 spike (S) protein, main protease (Mpro) enzyme, and RNA-dependent RNA polymerase (RdRp) enzyme), and therefore, has therapeutic potential against SARS-CoV-2.	[2,14]
6.	*Oligoporus tephroleucus*(Greyling bracket)	Oligoporin A	Oligoporin A shows antiplatelet activity by increasing the intracellular levels of both cAMP and cGMP in platelets and significantly repressing the collagen-induced ERK2 phosphorylation while diminishing the binding of fibrinogen to its cognate receptor, integrin IIb/IIIa, to exert its antiplatelet activity. Thus, this compound is a valuable candidate against COVID-induced thrombosis.	[51]

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
