# Peer review of "Therapeutic Potential of Bioactive Compounds from Edible Mushrooms to Attenuate SARS-CoV-2 Infection and Some Complications of Coronavirus Disease (COVID-19)"

_jof, 2023, doi:10.3390/jof9090897_

Round 1
Reviewer 1 Report
The authors have done an extensive literature review of the therapeutic effects of mushrooms against SARS-COV-2. Overall the review highlights the understudied area of mushroom bioactive compounds.
Kindly go through the following to improve the manuscript:
1) Please look out for typos throughout the paper, eg. "coronary bat virus" fourth line of the introduction.
2) In table no.3: Lentinan function is unclear, it says regulation of immune response by enhancing the functional role of neutrophils, macrophages, and mononuclear cells. Could you please elaborate on the immunoregulatory role?
Author Response
Reviewer 1:
The authors have done an extensive literature review of the therapeutic effects of mushrooms against SARS-COV-2. Overall the review highlights the understudied area of mushroom bioactive compounds.
Reply: Thank you for your encouraging words. We have addressed all the comments and corrected them in the revised manuscript.
Kindly go through the following to improve the manuscript:
Query 1: Please look out for typos throughout the paper, eg. "coronary bat virus" fourth line of the introduction.
Reply: Thank you for correcting us. This error has been fixed throughout the manuscript.
Query 2: In Table no.3: Lentinan function is unclear, it says regulation of immune response by enhancing the functional role of neutrophils, macrophages, and mononuclear cells. Could you please elaborate on the immunoregulatory role?
Reply: Thank you for your valuable suggestion. Now we have mentioned the function of Lentinan in the revised manuscript.
Reviewer 2 Report
The manuscript entitled: therapeutic potential of bioactive compounds from the edible mushrooms to attenuate SARS-COV-2 infection 2. I have no major suggestions and I believe this study is very interesting in the sense of emphasizing the importance of wild mushrooms in the treatment of Corona Virus. However, I suggest an improvement in the scientific and didactic quality of figures 1 and 5..Author Response
Reviewer 2:
The manuscript entitled: therapeutic potential of bioactive compounds from the edible mushrooms to attenuate SARS-COV-2 infection 2. I have no major suggestions and I believe this study is very interesting in the sense of emphasizing the importance of wild mushrooms in the treatment of Corona Virus.
Reply: Thank you for your appreciation and encouraging words.
Query 1: However, I suggest improving the scientific and didactic quality of Figures 1 and 5.
Reply: Thank you for your suggestions. The figures have been edited and improved.
Reviewer 3 Report
The review is interesting and comprehensive. Below I present some minor comments:
L 227 - They contain bioactive compounds such as penicillin, 227 griseofulvin, and the immunosuppressant cyclosporine A. - Does this sentence refer to mushrooms? As far as I know fungi are the source of antibiotics, however not mushrooms but microscopic organisims such as Penicillium. Please revised this part of the manuscript or provide proper references.
I think that the titles of the section 4.1. and 4.2. are too long and complicated.
L 240 - BRM instead BRD
L 431 - it should be NK T-cell
L 491 caju also in italic
Author Response
Reviewer 3:
The review is interesting and comprehensive. Below I present some minor comments:
Query 1: L 227 - They contain bioactive compounds such as penicillin, 227 griseofulvin, and the immunosuppressant cyclosporine A. - Does this sentence refer to mushrooms? As far as I know fungi are the source of antibiotics, however not mushrooms but microscopic organisms such as Penicillium. Please revised this part of the manuscript or provide proper references.
Reply: Thank you for correcting us. This mistake is now corrected in the revised manuscript.
Query 2: I think that the titles of section 4.1. and 4.2. are too long and complicated.
Reply: Thank you for your suggestion. We have rephrased the title of sections 4.1 and 4.2
Query 3: L 240 - BRM instead BRD
Reply: Corrected in the revised manuscript.
Query 4: L 431 - it should be NK T-cell
Reply: Corrected as suggested.
Query 5: L 491 caju also in italic
Reply: Corrected as suggested.
Reviewer 4 Report
Please see the attachment

Minor editing of English language required
Author Response
Reviewer 4:
Comments and Suggestions for Authors of the manuscript (ID jof-2474783) under the title "Therapeutic potential of bioactive compounds from the edible mushrooms to attenuate SARS- COV-2 infection and some complications in Coronavirus disease (COVID-19)" by Paran Baruah, Aparup Patra, Sagar Barge, Mojibur R Khan, Ashis Kumar Mukherjee
General characterization of the manuscript (MS).
The reviewed manuscript is devoted to an important problem of therapeutic strategies introduced to combat COVID-19 caused by SARS-CoV-2, a highly infectious positive RNA virus spread with increased mortality and morbidity. One of the strategies is to use natural products. This approach could implement mushrooms with their plentiful bioactive macromolecules proved to be a source of various therapeutic agents or drug prototypes.
Some specific comments
Query 1: The manuscript is relevant in the field, and addresses a specific gap in the field related to the mushrooms products application for combating COVID-19. However, to decide if the manuscript adds to the subject area compared with other published material, additional explanations are required. A rather large amount of recent reviews is concerned with the therapeutic potential of bioactive compounds from mushrooms to combat COVID-19 infection. Examples are as follows:
Shahzad, F., Anderson, D., & Najafzadeh, M. The antiviral, anti-inflammatory effects of natural medicinal herbs and mushrooms and SARS-CoV-2 infection. Nutrients 2020. 12(9), 2573.
Phillips, J. M., Ooi, S. L., & Pak, S. C. Health-promoting properties of medicinal mushrooms and their bioactive compounds for the COVID-19 era–an appraisal: do the pro-health claims measure up?. Molecules 2022. 27(7), 2302.
Mohiuddin, A. K. Can Medicinal Mushrooms Fight Against SARS-CoV-2/COVID-19?. Journal of Internal Medicine: Science & Art 2021. 2(1), 23 - 24. https://doi.org/10.36013/jimsa.v2i1.57
Galanakis, C. M., Aldawoud, T. M., Rizou, M., Rowan, N. J., & Ibrahim, S. A. Food ingredients and active compounds against the coronavirus disease (COVID-19) pandemic: A comprehensive review. Foods 2020. 9(11), 1701.
Barbosa, J. R., & de Carvalho Junior, R. N. Polysaccharides obtained from natural edible sources and their role in modulating the immune system: Biologically active potential that can be exploited against COVID-19. Trends in Food Science & Technology 2021. 108, 223-235.
Rahman, M. A., Rahman, M. S., Bashir, N. M. B., Mia, R., Hossain, A., Saha, S. K., ... & Sarker, N. C. Rationalization of mushroom-based preventive and therapeutic approaches to COVID-19. International journal of medicinal mushrooms 2021. 23(5).
Pu, Y., Chen, L., He, X., Ma, Y., Cao, J., & Jiang, W. Potential beneficial effects of functional components of edible plants on COVID-19: Based on their anti-inflammatory and inhibitory effect on SARS-CoV-2. Food Innovation and Advances 2023. 2(1), 44-59.
That is why the authors should specify the originality of the present review. A kind of "Novelty statement" should be presented within the framework of Introduction, which describes the originality of manuscript. The previous published reviews on the problem should be cited in this subsection and throughout the manuscript where appropriate. May be, new references have to be introduced in the discussion within the manuscript.
Reply: Thank you for your encouraging words and for giving your precise time for thorough revision to improve the quality of the manuscript. We have taken all your suggestions in a very positive spirit and tried our best to improve the quality of the manuscript. In the introduction section, we discussed how this review article adds to the subject area compared with other published material. Further, a novelty statement is also included in the introduction section.
Query 2: The section 2. "Structure and mechanism of infection" seems to be excessive in proportion to the whole manuscript. This section reports the information with no reference to the main question addressed by the research, i.e. the potential of bioactive compounds from the edible mushrooms.
Reply: Thank you for your suggestion. We have shortened section 2 and included proper references in the revised manuscript.
Query 3: The same (see point 2) is for the sections 3.2 "Antibody-based immunotherapeutic strategies", 3.3 "Cell therapy", and 3.4 "Nutraceutical supplementation".
Reply: Thank you for your valuable suggestions. In section 2, we have briefly discussed the currently available treatment strategies against COVID-19 and its limitation. However, as per your suggestion, we have shortened this section.
Query 4: In the sections 3.1 "Repurposed drugs", the drugs remdesivir, lopinavir/ritonavir and hydroxychloroquine are mentioned. Provided that these drugs are related in some aspect to mushroom-derived bioactive compounds, or may be another fungi, this data should be discussed to be closer to the main question addressed by the research.
Reply: In section 3.1, we have just mentioned the current treatment against COVID-19. However, per your suggestion, we have discussed some recent research on mushroom-derived bioactive compounds that can be used as a repurposed drug against COVID-19.
Other comments:
Query 5: Abstract, Line 19: "polysaccharides, proteins, and organic molecules have been used ...". Please correct, since polysaccharides and proteins are the organic molecules, too.
Reply: This sentence is corrected in the revised manuscript.
Query 6: Table 1, "Serial No. 11": "Lamda" should be replaced by "Lambda".
Reply: Corrected in the revised manuscript
Query 7: Line 214: "... Versicolor" should be replaced by "... versicolor".
Reply: Corrected in the revised manuscript
Query 8: Line 230: "A member of the Basidiomycetes division of Fungi, mushrooms are ". Please, take into account that mushrooms comprise both basidiomycetes and ascomycetes [Gargano et al., 2017; Venturella et al., 2019; Elkhateeb et al., 2022].
Reply: Thank you for correcting us. This classification has been corrected in the revised manuscript with proper citation.
Query 9: Lines 307, 309: "Laccase" should be replaced by 'laccase" to begin this enzyme name with lower-case letter.
Reply: We have checked thoroughly and corrected in the revised manuscript.
Query 10: Additional comments on the figures: Figures 3A (external remarks are visible), 3C, 3E (proportions distorted), 4D (external remarks and proportions), 4E, 4F (proportions distorted).
Reply: Thank you for correcting us. We have fixed the mistakes and redrawn the structures of compounds in the revised manuscript.
Query 11: Additional comments on the tables: Table 3 contains numerous repetitions in a columns "Function" and "Isolated From", especially relating to ref. 72 (in a rightmost column). The Table 3 design should be changed.
Reply: Thank you for your valuable suggestion to improve the quality of the manuscript. We have reframed and changed the table's design in the revised manuscript. The redundancy in the table is now removed.
Round 2
Reviewer 4 Report
I have checked the revised version. New text portions, directly relevant to the MS ideas and results, have appeared in many sections of MS, from the Abstract, main text, figures, to References.
A significant addition has been made to Introduction. Perfecting modifications have been made within the sections 3.3 and 3.4 among others, and to Figures 3 and 4.
Studies focused on repurposed drugs isolated from edible mushrooms have been described in the section 3.1. The revised Table 3 now reflects more clearly a list of bioactive compounds isolated from mushrooms in relation to their action against viral infections.
Minor editing of English language required